# Sustaining Organizational Outcomes in Manufacturing Firms: The Role of HRM and Occupational Health and Safety

**Ali Ateeq** [1], **Abd Al-Aziz Al-refaei** [2,3,*], **Mohammed Alzoraiki** [1], **Marwan Milhem** [1], **Ali Nasser Al-Tahitah** [4] **and Abdulhadi Ibrahim** [5]

[1]  Administrative Science Department, College of Administrative and Financial Science, Gulf University, Sanad 26489, Bahrain; dr.ali.ateeq@gulfuniversity.edu.bh (A.A.)

[2]  Research Management Centre, International Islamic University Malaysia (IIUM), Jalan Gombak, Kuala Lumpur 53100, Malaysia

[3]  Faculty of Administration & Economics, Shabwah University, Shabwah, Yemen

[4]  Faculty of Leadership and Management, University Science Islam Malaysia (USIM), Nilai 71800, Malaysia; altahitah.ali@gmail.com

[5]  College of Business Administration, A'Sharqiyah University, Ibra 400, Oman; abdulhadi.ibrahim@asu.edu.om

*  Correspondence: ah_alrefaei@yahoo.com

**Abstract:** While there is burgeoning interest in the influence of human resource management (HRM) on sustainability organizational outcomes (SOO), the intricate interplay with Occupational Health and Safety (OHS) has not yet been explored, particularly in manufacturing firms' contexts. Therefore, this study aims to probe the symbiotic relationship between HRM practices, OHS, and SOO, spotlighting manufacturing firms. Data collection was conducted by utilizing a cross-sectional survey, convenience sampling technique, and a web-based form among the 256 respondents from an industrial company (Balexco) in Bahrain. Structural Equation Modeling (SEM) by Smart-PLS was used to analyze the collected data. Our analysis highlighted a significant positive relationship between HRM, SOO, and OHS. Moreover, this study highlighted the dual impact of direct and indirect HRM on SOO, mediated by OHS. These multi-layered insights reinforce the assumption that a comprehensive approach to HRM, aligning performance aspirations with employee well-being, is instrumental in improving SOO. This study is a novel contribution to the literature because, by uncovering the intricate interplay of HRM, OHS, and SOO, practical implications and limitations were provided.

**Keywords:** sustainable organizational outcomes; human resource management; occupational health; health and safety; mental health and well-being; sustainable organizational development; workplace health and safety; health and safety compliance

## 1. Introduction

The role of HRM in sustaining organizational outcomes is crucial for the success and longevity of any organization. It is concerned with sustainable labor relations, employee well-being, and the overall quality of life within the organization to achieve sustainable outcomes [1]. HRM achieves short-term benefits but places great emphasis on achieving long-term organizational goals, ensuring that short-term and strategic goals are aligned with the overall values and vision of the organization [2]. Establishing alignment is essential to maintaining sustainable organizational effectiveness over the long term [3].

However, HRM plays a vital role in the functioning of organizations, therefore making it imperative to allocate resources towards their professional advancement and enhancement [4,5], in order to effectively achieve organizational goals and objectives [6]. Furthermore, enhancing organizational outcomes is contingent upon the institution's capacity to establish a comprehensive climate [7]. In order to thrive in the current highly competitive economic landscape, organizations must consistently seek opportunities to enhance the competitiveness of their staff [8,9], which consequently leads to improved productivity and

sustainable organizational outcomes. Nevertheless, in nations that value global competitiveness as a means to achieve their overarching national development objectives, there exists a significant need for a highly qualified workforce. Therefore, certain organizations may have challenges in maintaining long-term sustainability organizational outcomes (SOO) due to inadequate planning, insufficiently skilled personnel, or a lack of business acumen [10].

Numerous organizations have started recognizing the potential of sustainability organizational outcomes (SOO) in enhancing their competitive advantage and fostering innovation across various aspects such as processes, goods and services, markets, and business models [11]. However, the primary underlying assumption that guides the achievement of sustainable outcomes within organizations is that the quality of human resource management (HRM) practices plays a crucial role. This assumption posits that environmental factors or organizational processes do not solely determine sustainability but are intricately connected to recognizing the individual and distinct value that each human resource brings to an organization [12]. Hence, the notion that "individuals in good health contribute to the overall success of organizations" is supported by previous research [13]. Suppose organizations allocate resources towards human resource management strategies that foster a positive alignment between individuals and the organization, characterized by mutual trust and recognition. In that case, this endeavor will confer a competitive advantage. This is because employees who experience higher levels of well-being, safety, engagement, and job satisfaction are more inclined to exhibit enhanced productivity and efficiency [12]. In addition to delineating work tasks and responsibilities, Human Resource Management (HRM) encompasses the broader aspect of enhancing the general well-being of employees within the organizational context. The aforementioned elements include factors such as the equilibrium between work and personal life, initiatives promoting physical and mental well-being, and prospects for individual and occupational development [14].

There exists a clear correlation between the well-being of employees and the productivity and efficiency of an organization. HRM emphasizes establishing a conducive work environment that facilitates optimal performance [15,16], and the OHS of employees, hence fostering their motivation, engagement, and job satisfaction [17]. According to Pfeffer [17], these endeavors will need both scholarly inquiry and collective engagement. However, it is essential to acknowledge that the emphasis on constructing sustainable organizations should not just be limited to the physical environment but should also extend to the environment of society. The adverse impacts of corporate practices extend beyond the realm of the natural environment. Individuals are experiencing worse health outcomes and untimely mortality due to specific patterns of behavior within organizations. Therefore, despite establishing the relationship between HRM practices.

Many scholarly investigations primarily center on examining the direct relationship between Human Resource Management (HRM) and either sustainable organizational outcomes (SOO) or occupational health and safety. A smaller number of individuals may contemplate the simultaneous consideration or examine the interconnected impacts of both factors. The potential impact of human resource management (HRM) on sustainable outcomes, specifically occupational health and safety (OHS), may not have been thoroughly investigated. Examining how health safety standards contribute to sustainability, mainly when effectively managed through human resource management, can address this knowledge deficit [12]. The current body of literature may exhibit a deficiency in examining the challenges and opportunities manufacturing firms encounter in human resource management, sustainable outcomes, and occupational health and safety. Prior research may have primarily focused on Western or developed countries. Examining these associations within diverse cultural or economic settings, such as emerging markets or regions with distinct cultural characteristics, can provide novel perspectives. Therefore, the aim is to investigate the direct impact of HRM on SOO and occupational health safety and the indirect impact of HRM on sustainable organizational outcomes through the mediating role of occupational health safety in manufacturing firms.

## 2. Background of This Study

### 2.1. Challenge of Health and Safety Practices in the Manufacturing Industry

Industrial companies prioritize the safety and well-being of their employees, striving to maintain a secure work environment. However, like many organizations, industrial companies face various challenges in effectively implementing health and safety practices across their operations. This article explores the key challenges faced by the companies and examines the strategies and approaches adopted to address them. By analyzing these challenges, industrial companies aim to foster a safer workplace and promote a health and safety culture among their employees.

#### 2.1.1. Risk Assessment and Management

Risk assessment and management are fundamental aspects of ensuring a safe working environment. Identifying potential hazards and continuously monitoring workplace conditions require regular inspections and assessments [18]. Industrial companies acknowledge the significance of risk management and conduct frequent risk assessments to mitigate potential risks. To achieve continuous improvement, the company actively encourages employees to report potential hazards and fosters a culture of constant improvement in safety practices [19].

#### 2.1.2. Staff Training and Involvement

Employee training and engagement are essential components of maintaining a safe workplace. Industrial companies recognize the importance of providing comprehensive training to their workforce to ensure that employees are well-informed about health and safety policies [20]. To address the challenge of universal employee participation in safety practices, industrial companies organize interactive training sessions and open forums and promote a sense of ownership among employees in maintaining a secure work environment [21]. By involving staff in safety-related decisions and encouraging feedback, the company aims to instill a strong safety culture.

#### 2.1.3. Communication

Effective communication is the cornerstone of a successful health and safety program [22]. Industrial companies understand the significance of transparent and timely communication to keep employees informed about potential hazards, emergency procedures, and best practices. Industrial companies are committed to enhancing communication channels, conducting regular safety meetings, and leveraging technology for seamless information dissemination (occupational health and safety administrative) to bridge the gap between employees and supervisors. By encouraging an open-door policy and fostering a culture of two-way communication, the company aims to promote a safer working environment [23].

#### 2.1.4. Mental Health and Well-Being

Recognizing the growing prominence of mental health issues, industrial companies actively support the mental wellness of their workforce [24]. The companies have implemented various initiatives, including access to counseling services, wellness programs, and awareness campaigns to reduce stigma [24]. By addressing stress, anxiety, and other mental health concerns, industrial companies aim to create a positive work environment and enhance employee well-being.

#### 2.1.5. Technological Developments

The rapid advancement of technology poses both opportunities and challenges for industrial companies concerning health and safety practices [25]. Embracing automation, artificial intelligence, and remote working can improve safety and efficiency. However, it also introduces new risks that demand careful management. Industrial companies remain dedicated to staying updated with technological advancements and continuously

reassessing safety protocols to align with these changes [26]. Additionally, the company invests in training its employees to adapt to evolving technologies safely. Industrial companies actively addressed health and safety challenges through risk management, staff training, mental health support, and technological advancements. Their unwavering commitment to employee well-being sets a standard for industry-wide safety practices.

## 3. Literature Review and Hypothesis Development

### 3.1. HRM Practices on Sustainable Organizational Outcomes (SOO)

The concept of sustainable organizational performance has garnered considerable interest in recent times, primarily due to its enduring impact on both business prosperity and societal welfare. Epstein and Buhovac [26] argue that contemporary human resource management (HRM) practices have incorporated sustainability indicators into their performance measurement systems. This integration allows organizations to effectively monitor and enhance their sustainable outcomes. Ref. [27] comes to the conclusion that an organization should enhance its practices and capabilities in order to establish a comprehensive set of values that are geared towards fostering the growth of employees' skills and abilities, which will ultimately contribute to the attainment of organizational objectives in both the short and long term, thereby ensuring the organization's sustainability. Therefore, scholars place significant emphasis on the congruence between HRM practices and sustainability objectives. According to [28], the integration of sustainability considerations into HRM processes is advocated. This is because the HRM function possesses a distinctive position that enables it to support the development and implementation of sustainability strategies. Moreover, this integration has a positive influence on the organization's capacity to attain sustainable performance outcomes [29].

Another study highlights the positive impact of green human resource management (HRM) on the sustainable performance of healthcare organizations. The study was conducted by Mousa and Othman [30]. The findings of this study indicated that although the level of green HRM implementation was moderate, the overall sustainable organizational performance was found to be high. According to Acquah et al. [31], implementing HRM practices that foster knowledge sharing, learning, and cross-functional collaboration can lead to innovative solutions for challenges, positively influencing SOO. According to Ren et al. [32], it is crucial to highlight the significance of human resource management (HRM) practices that align with sustainability objectives. These practices, including green recruitment and employee training, have yielded favorable outcomes for organizations regarding sustainability. A study conducted by Bakker and Demerouti [33] found that when employees are actively involved in organizational practices, there is a higher probability of their actions being in line with the organization's sustainability objectives. This alignment subsequently leads to improved levels of efficiency and effectiveness. Based on previous discussions, the current study assumes a positive effect of HRM practices on SOO. Therefore:

**H1.** *There is a positive relationship between Human Resource Management (HRM) and sustainable organizational outcomes in manufacturing firms.*

### 3.2. The Relationship between Human Resource Management and Occupational Health and Safety

Effective HRM practices' impact on OHS is widely acknowledged in the academic literature and various industry reports. For example, the International Labour Organization (ILO) emphasizes the importance of integrating OHS policies into HRM practices to create safe and healthy workplaces [International Labour Organization, "Promotional Framework for Occupational Safety and Health Convention, 2006 (No. 187)"] [34]. Similarly, numerous research articles and studies discuss the relationship between OHS and HRM. One such study by Singh et al. [35] emphasizes the role of HRM in creating a safety culture and reducing workplace accidents, and another study conducted by Naji et al. [36] is entitled "The mediating role of safety culture and climate in the relationship between HRM practices and accidents at work".

Regarding the positive impact of OHS on HRM, research conducted by the European Agency for Safety and Health at Work (EU-OSHA) shows that organizations with strong OHS practices experience improved employee well-being, job satisfaction, and reduced turnover rates [source: European Agency for Safety and Health at Work (EU-OSHA), "Promoting a positive culture of health and safety at work" [37]. Various experts and institutions have advocated a collaborative approach between HRM and OHS. For instance, the Canadian Centre for Occupational Health and Safety (CCOHS) highlights the significance of integrating OHS into HRM functions to foster a safer work environment [Canadian Centre for Occupational Health and Safety (CCOHS) [38]. Therefore, in conclusion, while the specific manufacturing companies may not have been directly cited due to the limitations of the knowledge cutoff, the integration of HRM and OHS and their positive impact on workplace safety and employee well-being is supported by various reputable sources in the field of occupational health and safety. In light of the aforementioned theoretical underpinnings and empirical findings, the following hypotheses

**H2.** *Human Resource Management (HRM) Positively Influences Occupational Health and Safety (OHS) in Manufacturing Firms.*

### 3.3. Occupational Health and Safety (OHS) and Organizational Outcomes

In today's world, human resource managers have more significant challenges pertaining to occupational health and safety in comparison to previous periods. The rationale behind this is that workers, similar to other resources, require upkeep and attention to optimize their production [39]. In light of these circumstances, it is important to recognize that health and safety should not be regarded as an isolated function or obligation but rather as a comprehensive endeavor that seeks to enhance the productivity, profitability, and competitiveness of a company [40]. Hence, it is important to prioritize the provision of a hazard-free work environment for employees, as research indicates that individuals working in such conditions exhibit higher levels of productivity. However, corporations that take care of their employees as a part of their responsibility will improve their performance [41,42].

According to the study conducted by Bell et al. [43], it was observed that British organizations that had superior noise control measures tended to exhibit characteristics such as bigger size, more occupational health and safety (OHS) expertise, stronger OHS values, and a higher propensity to allocate resources towards OHS expenditures. These factors were shown to contribute to the achievement of sustainable outcomes. The study undertaken by Yanar et al. [44] used a qualitative case study approach to examine the impact of an organization's occupational health and safety (OHS) performance. The research revealed that the cross-case analysis identified three separate categories of organizations based on their overall occupational health and safety (OHS) performance: high, medium, and poor. There is a positive correlation between higher company ratings, which serve as a marker for organizational success, and improved occupational health and safety (OHS) performance within the workplace.

Gopang et al. [45] conducted an empirical investigation to ascertain the correlation between occupational health and safety measures and the performance of 35 SMEs in Pakistan. The findings indicated a statistically significant, moderately positive association between occupational health and safety and performance. This observation indicates that the implementation of Occupational Health and Safety Measures was inadequate, resulting in an impact on the operational effectiveness of SMEs. The authors reached the conclusion that it is imperative for SMEs to prioritize the effective execution of appropriate strategies. The research conducted by Kaaria and Mwaruta [46] shows that Occupational Safety and Health (OSH) training substantially impacts the performance of cement manufacturing companies in Kenya. This finding provides more evidence that the influence of the predictor factors on the outcome variables is contingent upon the level of support offered by management.

However, in highly competitive contexts, the integration of occupational health and safety (OHS) measures is crucial for ensuring sustainable organizational outcomes (SOO),

sustainable growth, and the enduring viability of industrial organizations. The implementation of health and safety measures by industry organizations has been found to be inadequate, mostly due to a lack of attention and prioritization by industrialists towards this crucial aspect [45]. Therefore, the current study examines the effect of OHS on organizational sustainable outcomes in manufacturing organizations and assumes that:

**H3.** *Occupational Health and Safety (OHS) has a positive effect on organizational sustainable outcomes (SOO) in manufacturing firms.*

**H4.** *Occupational Health and Safety (OHS) mediating relationship between Human Resource Management (HRM) and sustainable organization outcomes (SOO) in manufacturing firms.*

In the current study, there are three constructs, including in the conceptual framework: the independent variable is HRM practices, while the dependent variable is sustainable organizational outcomes (SOO), and the mediating variable is occupational health and safety (see Figure 1).

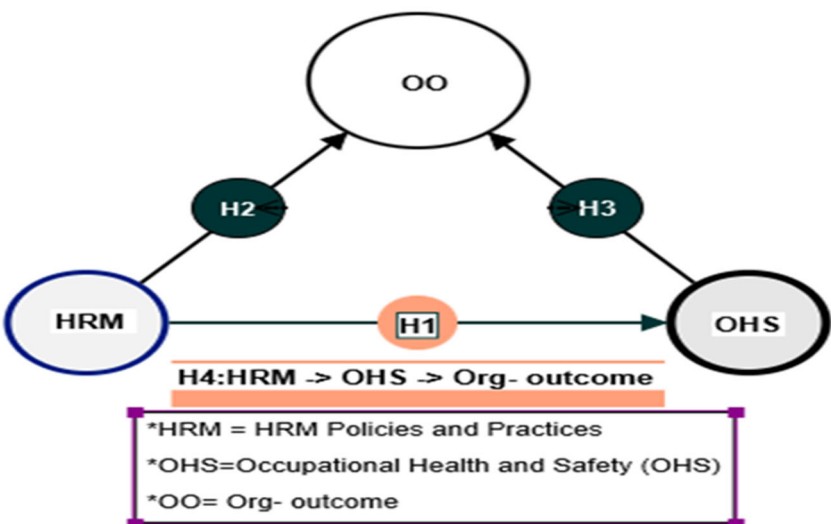

**Figure 1.** Author's Own Conceptual Framework and Hypotheses Model.

## 4. Methodology

### 4.1. Ethical Consideration

The research was conducted in compliance with ethical guidelines. Informed consent was obtained from all participants, ensuring that they were aware of this study's purpose and that their participation was voluntary [47,48]. Confidentiality and anonymity were maintained throughout this study.

### 4.2. Sample

Several employees of Balexco Company in Sitra, Kingdom of Bahrain were requested to participate in an online study regarding Examining the Influence of Human Resource Management Roles on Organizational Outcomes in Balexco Company: The Mediating Effect of Occupational Health and Safety. A total of 156 respondents were used as shown in Figure 2. A convenience sample of employees was contacted to participate in this research. A sample size of *n* = 156 was considered sufficient for this study, according to the table of Krejcie and Morgan 1970 [49]. In addition, a convenience sample was contacted to participate in this study.

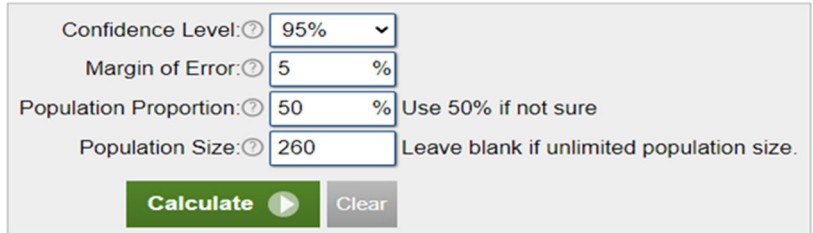

**Figure 2.** Sample Size Calculator.

*4.3. Survey*

A cross-sectional survey was designed to assess the impact of A cross-sectional survey was designed to assess the Influence of Human Resource Management Roles on Organizational Outcomes with Mediating Effect of Occupational Health and Safety. An online survey was conducted. A web-based survey was created using Google Forms and emailed to the employees of the company. Data were collected between the 13 June 2023 and 14 July 2023.

*4.4. Data Analysis*

This study employed descriptive analysis through SPSS version 28 to interpret the data, utilizing partial least squares SEM (PLS-SEM) for estimating complex cause-and-effect relationships between latent variables. The internal consistency of both exogenous and endogenous variables was assessed using Cronbach's alpha, resulting in a score of 0.854. This score implies a high level of consistency within the questionnaires, validating the reliability of the instrument used in this study [50].

**5. Results**

*5.1. Background Characteristics of the Respondents*

Table 1 provides an overview of the respondents' profiles in this study. There were a total of 154 respondents. Among them, 143 (92.8%) identified as male, and 11 (7.2%) were female. In terms of the ages, 42 (27.2%) were below 30, while 69 (44.8%) fell within the age range of 31–40, 26 (16.8%) were between 41–50, and 17 (11%) were above the age range of 50. Regarding experience, out of the total respondents, 22 (14.2%) have less than 10 years of experience, while 132 (85.7%) have more than 10 years of experience. In terms of education, 82 (53.2%) have a high school education, 37 (24%) have a Diploma, 18 (11.5%) have a Bachelor's degree, and 17 (11%) fall into the "Others" category for education.

**Table 1.** Respondent's profiles.

| Characteristics | Categories | *n* | Percentage (%) |
|---|---|---|---|
| Gender | Male | 143 | 92.8 |
| | Female | 11 | 7.2 |
| Age | Less than 30 | 42 | 27.2 |
| | 31–40 | 69 | 44.8 |
| | 41–50 | 26 | 16.8 |
| | above 50 | 17 | 11 |
| Experience | <10 | 22 | 14.2 |
| | >10 | 132 | 85.7 |
| Education | High school | 82 | 53.2 |
| | Diploma | 37 | 24 |
| | Bachelor | 18 | 11.5 |
| | Others | 17 | 11 |

Table 2 summarizes a research study's key statistics and rankings for three different constructs (HRM Policies and Practices, Occupational Health and Safety, and Sustainable Organizational Outcome). It provides information about the number of items, mean values, standard deviations, rankings for each construct, and an overall average for all constructs.

**Table 2.** μ ± SD and Rank for the Variables.

| Constructs | Number of Items | Code | Mean ± SD | Rank |
|---|---|---|---|---|
| HRM Policies and Practices | 4 | HRM | 0.858 ± 0.016 | 1 |
| Occupational Health and Safety (OHS) | 5 | OHS | 0.776 ± 0.024 | 2 |
| Sustainable Organizational Outcome | 7 | OO | 0.714 ± 0.035 | 3 |
| Average | 16 | | 0.783 ± 0.025 | |

Note: Rating scales: 5-point Likert scale (1) Strongly Disagree, (2) Disagree, (3) Natural, (4) Agree, and (5) Strongly Agree, Scores range from 1 to 5, with higher scores indicating better competence. SD: Standard Deviation [48].

This study demonstrates a pronounced understanding among respondents regarding the influence of Human Resource Management (HRM) Policies and Practices on Organizational Outcome (OO). All observed variables yielded scores exceeding 0.783, with HRM emerging as the most significantly impacted factor, registering a high mean score of 0.858 with a standard deviation of ±0.016. Following HRM, Occupational Health and Safety (OHS) was noted with a mean score of 0.776 and a standard deviation of ±0.026. On the other hand, organizational Outcome (OO) exhibited the lowest mean of 0.714, accompanied by the highest standard deviation of ±0.035. The collective mean score was determined to be 0.783 with a standard deviation of ±0.025, reflecting substantial variability within the data. This reinforces the implication that HRM Policies and Practices play a critical role in shaping Organizational Outcomes. A bar graph in Figure 3 illustrates these results.

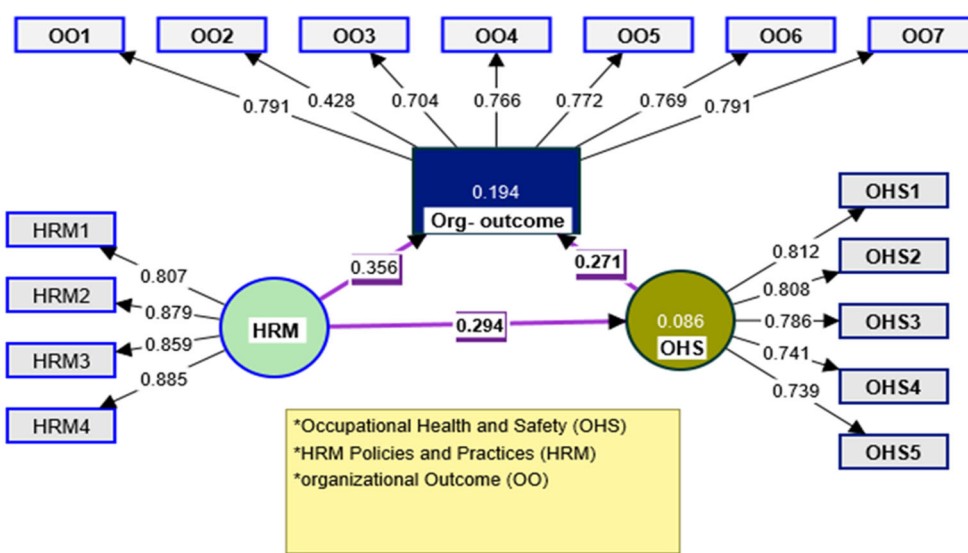

**Figure 3.** Measurement model of the study.

*5.2. Reliability and Composite Reliability*

In this study, SPSS version 28 was employed to evaluate the instrument's reliability, yielding satisfactory Cronbach's alpha (0.881 to 0.837) and Composite Reliability (CR) values (0.890 to 0.840) as shown in Table 3, according to D'Cruz. Du discriminant validity was confirmed as the square root of the Average Variance Extracted [51] surpassed the correlation between the latent constructs [52,53] with all CR estimates exceeding 0.880 and

Cronbach's alphas over 0.7. The findings underscored significant reliability and validity, as detailed in Table 3.

**Table 3.** Reliability and Composite Reliability.

| Construct | Cronbach's Alpha | Composite Reliability | Average Variance Extracted |
|---|---|---|---|
| HRM | 0.881 | 0.890 | 0.737 |
| OHS | 0.837 | 0.844 | 0.605 |
| Org- outcome | 0.845 | 0.840 | 0.529 |

"Cronbach's alpha is an average measure of internal consistency and item reliability and is preferred when EFA is used for factor extraction. <0.7 is accepted. CR: measure scale reliability overall and preferred with CFA. AVE: measures the level of variance captured by a construct 0.5 accepted". AVE: Average Variance Extracted. CR: Composite Reliability [54].

### 5.3. Convergent Validity

In this research, convergent validity was assessed according to specific criteria, including factor loadings, composite reliability, and average variance extracted (AVE), as defined by Naji et al. [55] and shown in Table 4. All factor loadings were above the 0.50 threshold, and composite reliability values ranged from 0.890 to 0.840, exceeding the recommended 0.7. AVE values were also above the 0.5 cutoff. These results demonstrate that the measurement model used in this study exhibits strong and robust convergent validity.

**Table 4.** The Convergent Validity Analysis.

| Construct | Code | Number of Items | Factor | Construct | Code |
|---|---|---|---|---|---|
| HRM | HRM | 4 | 0.858 | 0.890 | 0.737 |
| OHS | OHS | 5 | 0.777 | OHS | OHS |
| Sustainable Org- outcome | OO | 7 | 0.717 | 0.840 | 0.529 |

Key: "factor loading: variance explained by the variable on that particular factor <0.7 or higher to be accepted [56]. CR: measure scale reliability overall, preferred with CFA. AVE: measures the level of variance captured by a construct 0.5 accepted". AVE: Average Variance Extracted. CR: Composite Reliability.

### 5.4. Assessment of Measurement Model

The measurement model of the current study contains three constraints, HRM policies and practices (HRM), Occupational Health and Safety (OHS), and organizational Outcome. Assessment of the measurement model for convergent and discriminant validity.

### 5.5. Discriminant Validity for Latent Variables

In the present study, the discriminant validity between constructs was ascertained by employing the Fornell and Larcker (1981) criterion [57]. This statistical examination affirmed that, although the constructs under investigation are interrelated, they indeed quantify discrete concepts. This conclusion is not only consistent with the current findings but is also corroborated by preceding research [58,59]. A detailed representation of the Discriminant Validity Analysis can be found in Table 5.

**Table 5.** Discriminant Validity Analysis.

| | HRM | OHS | Org- Outcome |
|---|---|---|---|
| HRM | 0.858 | | |
| OHS | 0.294 | 0.778 | |
| Org- outcome | 0.356 | 0.353 | 0.727 |

Note: The square root of the average variance extracted is represented by a diagonal, while the other elements reflect the correlation estimate.

*5.6. The Prediction Relevance of the Model*

In the context of multivariate data analysis, the determination coefficient $R^2$, denoting the proportion of the variance in a specific endogenous variable that is predictable from the independent variables, serves as an essential metric for assessing model performance [42,60]. The validity of the current model was substantiated through the utilization of the sample reuse method. An exemplary fit was discerned, as evidenced by an $R^2$ value of 18% for the dependent variable, Organizational Outcome, amongst Balexco employees in Bahrain. Additionally, the Occupational Health and Safety (OHS) variable yielded an $R^2$ of 19% within the same organization. The proximity of these values implies a significant contribution of the independent variables to the variance in employee performance within the given context. Strikingly, over 18% variability in job performance was observed during this study period, further corroborating the robustness of the established relationships. The $R^2$ of these variables is systematically delineated in Table 6 and visually represented in Figure 4, providing a comprehensive understanding of the impact of the predictors on the dependent variables in question.

**Table 6.** Coefficient of determination result: $R^2$.

|  | R-Square | R-Square Adjusted |
|---|---|---|
| OHS | 0.086 | 0.083 |
| Org- outcome | 0.194 | 0.188 |

Key: OHS = Occupational Health and Safety (OHS), Org- outcome = Sustainable Organizational Outcome, Higher value is preferred: 0.67 substantial, 0.33 average, 0.19 weak [61].

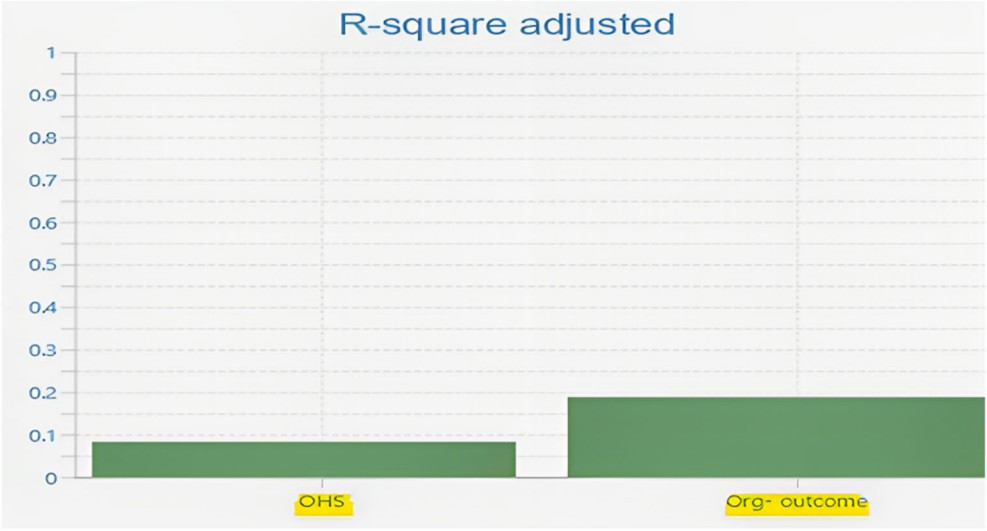

**Figure 4.** $R^2$. $R^2 = 1 - Rss/TSS$.

*5.7. Effect Size*

In the present study, we employed effect size as a measure to scrutinize the relationships between various variables. Following classification of effect sizes were segregated into small (0.02), medium (0.15), and large (0.35) categories. Table 7 reveals that among the factors analyzed, Occupational Health and Safety (OHS) manifested the most minimal impact on organizational outcomes, as delineated in reference [54]. In contrast, Inclusive Human Resource Management (HRM) emerged as having the most profound influence on OHS. Intriguingly, the interaction between HRM and OHS was found to account for approximately 86% of the variance in organizational outcomes. A comprehensive representation of the effect sizes corresponding to these variables is available in Table 7 and Figure 5.

**Table 7.** f-square.

| | f-Square |
|---|---|
| HRM → OHS | 0.094 |
| HRM → Org- outcome | 0.087 |
| OHS → Org- outcome | 0.084 |

Note: $f^2$ = ($R^2$ included − $R^2$ excluded)/(1 − $R^2$ included). Key: $f^2$ 0.02 weak, 0.15 moderate, 0.35 strong effects [62].

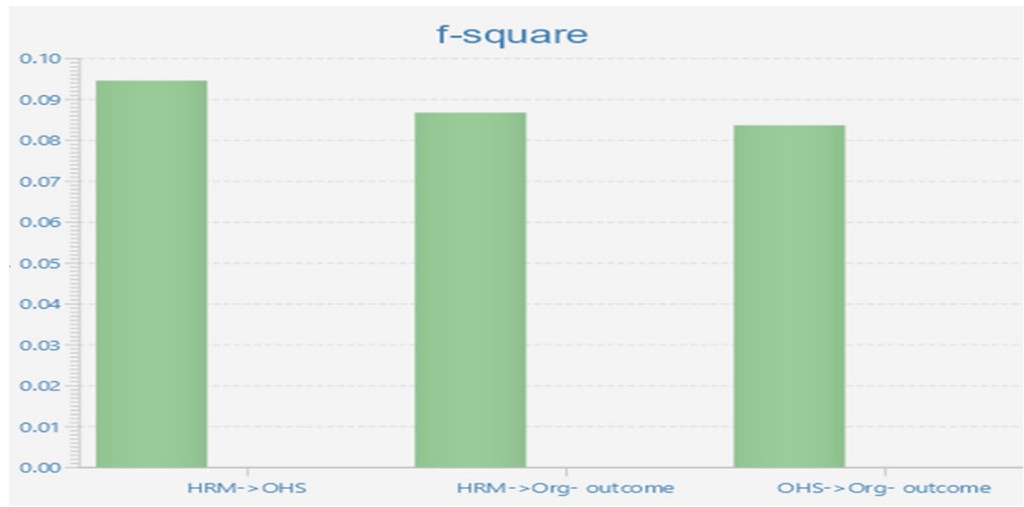

**Figure 5.** Effect size $f^2$. Effect size (f) = $\sqrt{(\eta^2 / (1 - \eta^2))}$.

*5.8. The Assessment of the Inner Model and Hypotheses Testing Procedures*

The result hypotheses of the study are shown in Table 8. This reveals the direct effect of Human Resource Management policies and practices (HRM) on Organizational Outcomes, occupational health and safety (OHS), and the direct effect of occupational health and safety (OHS) on Organizational Outcomes.

**Table 8.** Mean, STDEV, T values, *p* values, Decision.

| NO | Hypothesis | β | μ | SD | T. Value | *p* Values | Decision |
|---|---|---|---|---|---|---|---|
| H1 | HRM → OHS | 0.294 | 0.299 | 0.060 | 4.889 | 0.000 | Supported |
| H2 | HRM → Org- outcome | 0.276 | 0.279 | 0.058 | 4.728 | 0.000 | Supported |
| H3 | OHS → Org- outcome | 0.271 | 0.277 | 0.059 | 4.623 | 0.000 | Supported |

Note: "HRM = Human Resource Management Org = Organizational Outcomes, and OHS = Occupational Health & Safety" Beta (β); Values from −1 to +1. Assess the significance and confidence intervals. *p*-values; Significance value is based on the degrees of freedom $p < 0.05$ Cheah [61]".

*5.9. Path Model Significance Results*

In the present study, the bootstrapping method was employed in conjunction with Smart PLS4 to ensure that the path coefficients were statistically significant. Specifically, bootstrapping was used to generate t-values corresponding to each path coefficient. As a corollary, *p*-values for the hypotheses were also produced, as delineated in Table 6. With respect to the relationship between Human Resource Management (HRM) Policies and Practices and Occupational Health and Safety (OHS), the analysis revealed a positive impact on OHS as the dependent variable. This relationship was found to be statistically significant at the 0.01 level (β = 0.294, μ = 0.299, SD = 0.060, t = 4.889, $p < 0.000$). The support for this relationship was also indicated by the t-value exceeding the threshold of 1.96, combined with a low standard deviation (SD) of 0.060. Similarly, HRM Policies and Practices (HRM) were found to have a positive impact on Organizational Outcome as the dependent variable (HRM → Org- outcome), also significant at the 0.01 level (β = 0.276, μ =

0.279, SD = 0.058, t = 4.728, *p* < 0.000). Consequently, the alternate hypotheses were accepted and the null hypotheses were rejected, reflecting the significant results for this variable. Furthermore, the relationship between Occupational Health and Safety and Organizational Outcome (OHS → Org- outcome) was supported at the 0.01 level of significance (β = 0.271, μ = 0.277, SD = 0.059, t = 4.623, *p* < 0.000). This particular variable demonstrated the strongest support among respondents, as evidenced by the significant *p*-value and the lowest standard deviation, signifying less dispersion among the participants. The PLS4 bootstrapping approach facilitated the analysis of all the aforementioned relationships. The results are encapsulated graphically in Figure 6, visually representing the hypotheses and their statistical validation. The findings collectively underscore the vital influence of HRM Policies and Practices on Occupational Health and Safety and Organizational Outcomes, shedding new light on the underlying mechanisms that govern these relationships in this study context.

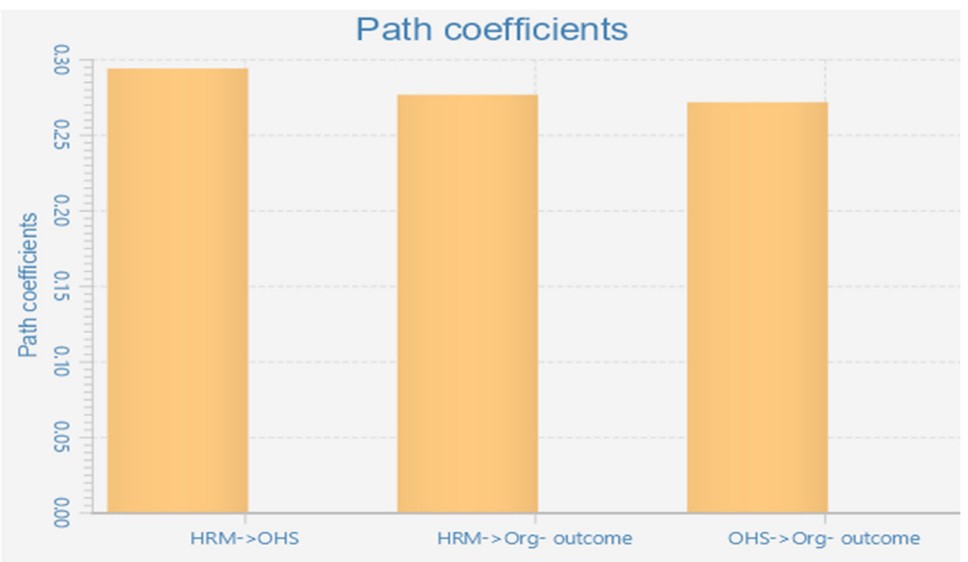

**Figure 6.** Path Coefficient Bar Chart.

*5.10. Testing the Level of Significance of the Indirect Effect*

Figure 6 illustrates the relationship among Human Resource Management (HRM), Occupational Health and Safety (OHS), and organizational outcome (Org- outcome) within Balexco's employee structure. Specifically, the indirect impact of HRM on Org- Outcome through OHS is quantified as 0.08 (0.276 × 0.294), while the direct effect is 0.271.

In the context of Balexco's employee framework, OHS operates as a mediator connecting HRM with Org- Outcome. Since the direct impact is substantial but remains less than 80%, it is consistent with the parameters for partial mediation. Consequently, the indirect hypothesis is statistically regarded as evidencing partial mediation. A comprehensive presentation of the outcomes derived from the indirect analysis can be found in Table 9.

**Table 9.** A comprehensive presentation of the outcomes.

| | Original Sample (O) | Sample Mean (μ) | Standard Deviation (STDEV) | T Statistics (\|O/STDEV\|) | *p* Values |
|---|---|---|---|---|---|
| HRM → OHS → Org- outcome | 0.08 | 0.083 | 0.026 | 3.082 | 0.002 |

## 6. Discussion

The current study's findings highlight the dominant discourse in scholarly literature, which supports the notion that HRM practices have a beneficial impact on sustainable organizational outcomes. The validation of the initial hypothesis illustrates the significant impact of HRM on influencing and improving sustainable performance measures within manufacturing firms. The result obtained by data analysis confirmed that HRM practices have a positive and significant effect on sustainable organizational outcomes in manufacturing firms. The current finding indicated that it becomes apparent that contemporary HRM practices, which have incorporated sustainability metrics, have assumed a broader responsibility in safeguarding the well-being of employees and the wider social environment; this was consistent with the findings of [43]. The results derived from this study align with the perspective of [44], emphasizing the imperative for organizations to enhance their practices and capabilities. The phenomenon above contributes to the development of employees and serves as a foundation for the broader goals of the organization, which are essential for long-term, sustainable outcomes. The findings reiterate that the HRM field's unique role within organizations allows it to support the development and implementation of sustainability strategies. This finding aligns with the viewpoints expressed by [45,46]. As demonstrated by the present study, the importance of integration cannot be overstated to attain the desired sustainable outcomes. Additional comparisons with existing studies in the field, such as the research conducted by Mousa and Othman [30], provide indications regarding the degree to which HRM can influence sustainable outcomes. Ref. [47] observed that green HRM practices are moderately implemented in healthcare organizations. However, the significant levels of sustainable performance highlight the considerable impact of HRMs indirect influence on organizational outcomes. Furthermore, the current findings align with the research conducted by [48], highlighting the significance of HRM practices centered on knowledge sharing and collaboration in fostering innovative solutions. Addressing contemporary organizational challenges is of utmost importance, as it strengthens the concept of sustainability. In addition, the study conducted by Bakker and Demerouti [50] on the relationship between employee involvement and organizational sustainability objectives aligns with our findings. The current research study provides additional support for the assertion that employees, when aligned with human resource management practices, frequently demonstrate behaviors that promote organizational sustainability, leading to improved efficiency and effectiveness. However, the results of our study emphasize the favorable influence of HRM practices on the maintenance of occupational health and safety (OHS) in the manufacturing sector. These findings align with the perspectives expressed in numerous scholarly articles. The significance of HRM practices in influencing occupational health and safety (OHS) outcomes is unquestionable, especially in manufacturing. The current finding is linked with the statement substantiated by the International Labour Organization (ILO), which emphasizes integrating OHS policies with HRM practices. This comprehensive approach ensures the implementation of safety protocols and facilitates establishing a work environment that promotes employee well-being and health. The current findings and perspective are consistent with previous studies, such as the study conducted by [51], which supports the effortless incorporation of both disciplines. The current study's finding provides further insight into the correlation between HRM and OHS, which is consistent with the study conducted by [52], which reveals the significant impact of HRM in cultivating a safety-oriented culture. The current result indicates that implementing safety measures through HRM practices substantially reduces workplace accidents. The position is additionally supported by [36], who demonstrate the crucial significance of safety culture and climate in facilitating the connection between HRM initiatives and measurable decreases in workplace accidents. The results of their study highlight the complex relationship between HRM practices and OHS outcomes. The current finding indicates that the relationship between HRM and OHS is not unidirectional but rather characterized by reciprocity. The implementation of HRM practices contributes to the enhancement of OHS standards. In turn, the establishment of strong OHS frameworks

reinforces the outcomes of HRM. Furthermore, the consensus among industry experts and prestigious institutions regarding the need for a cohesive integration of Human Resource Management (HRM) and Occupational Health and Safety (OHS) further reinforces the core findings of our study. Furthermore, the findings of this study indicate a clear and positive correlation between occupational health and safety (OHS) and sustainable organizational outcomes within the manufacturing sector. The current result showed that individuals in the present era encounter intricate occupational health and safety difficulties. This vision aligns consistently with prior scholarly research, advocating for the thorough integration of occupational health and safety (OHS) within the organizational structure. It highlights the crucial role of OHS in improving overall company productivity and profitability [56]. Alsamawi, Darun, and Panigrahi [41] highlighted the significance of organizations prioritizing employee welfare, stating that such organizations fulfill their societal responsibility and improve their overall performance.

The results of the current study indicate that there are a variety of factors that contribute to achieving sustainable results. These include various attributes, including resource allocation and a firm commitment to Occupational Health and Safety (OHS) values. There is a clear relationship between higher organizational ratings and improved occupational health and safety (OHS) performance. This insight is consistent with previous studies such as [62,63]. The survey by [64] provided additional support for the above association, highlighting the existence of a moderately positive relationship between occupational health and safety (OHS) measures and operational efficiency within small and medium-sized enterprises (SMEs). These results provide evidence of the overall importance of occupational health and safety (OHS) in manufacturing settings. Ref. [65] conducted a study on cement manufacturing companies in Kenya, which supports the above view by highlighting the significant impact of OSH training on the overall performance of these companies.

### 6.1. Theoretical Implication

The theoretical contribution of this study is that it uncovers the intricate relationship between human resource management (HRM) practices, Occupational Health and Safety (OHS), and sustainability organizational outcomes (SOO) within the context of manufacturing firms. This study is unique as it not only confirms a positive correlation between HRM, OHS, and SOO but also reveals the dual impact of HRM on SOO, both directly and indirectly through the mediation of OHS. Such insights emphasize the importance of a holistic HRM approach that integrates performance goals with employee well-being to enhance sustainability outcomes. This study fills a gap in the literature by bridging HRM practices, OHS considerations, and sustainability outcomes.

### 6.2. Practical Implication

Manufacturing companies must give priority to the integration of HRM practices that not only emphasize performance but also include Occupational Health and Safety (OHS) considerations. Through this action, individuals can substantially impact the achievement of sustainable objectives within an organization. Industrial enterprises must recognize the symbiotic connection between human resource management (HRM) practices and occupational health and safety (OHS). Enhancing employee well-being through occupational health and safety (OHS) initiatives can indirectly positively impact sustainability outcomes, as it fosters the development of a healthy and highly efficient workforce [14]. Managers in manufacturing firms should understand the dual influence of Human Resource Management (HRM) when making strategic decisions, encompassing both its direct and indirect effects mediated by Occupational Health and Safety (OHS). The achievement of their sustainability initiatives is intricately linked to this comprehension. Manufacturing enterprises ought to contemplate allocating resources towards occupational health and safety (OHS) measures, not solely as a compulsory regulatory requirement; however, as a strategic instrument that indirectly impacts the attainment of sustainable organizational outcomes. Implementing appropriate training programs, utilizing suitable tools, and adher-

ing to standardized procedures aimed at ensuring safety can result in a more motivated and committed workforce. Finally, by employing the PLS4 bootstrapping approach, our study adds robustness to these relationships. While earlier studies [64,66] employed other statistical techniques, our methodology allows for a more nuanced understanding, especially with the graphical representation provided.

## 7. Limitations and Future Research Directions

This study, while groundbreaking in its insights, is not without limitations. Firstly, the reliance on convenience sampling and a single industrial company from Bahrain might limit the generalizability of the findings across diverse manufacturing contexts. The cross-sectional nature of the data collection offers a snapshot in time, potentially overlooking longitudinal effects. There may also be unexplored confounding variables that might influence the relationships observed. Given these constraints, future research should consider employing random sampling techniques, diversifying the industrial sectors and geographic locations studied, and adopting longitudinal methodologies. Additionally, future endeavors could probe deeper into other potential mediators and moderators that might influence the relationship between HRM practices, OHS, and SOO, thereby enriching the academic discourse and practical applications in the field.

## 8. Conclusions

The complex and influential relationship between Human Resource Management (HRM) practices, Occupational Health and Safety (OHS), and sustainable organizational outcomes in manufacturing firms is undeniable. Our research extensively examines the connection between HRM and sustainable organizational outcomes. We identify the direct impact of HRM and explore its indirect influence through the mediation of OHS. However, this study addresses a significant void in the scholarly literature by emphasizing the combined effects of HRM, both directly and through OHS mediation. It provides practical recommendations for industrial organizations seeking long-term and sustainable development. The statement suggests that HRM and OHS should not be considered separate entities but interconnected components that contribute to the long-term success of an organization. In light of the contemporary challenges faced by manufacturing sectors globally, the integration of the findings from this study may serve as a guiding principle, leading them toward achieving a balance between productivity, employee welfare, and sustainability.

**Author Contributions:** Conceptualization, A.A. and A.A.-A.A.-r. methodology, M.A.; software, M.M.; validation, A.N.A.-T. and A.I.; formal analysis, M.A.; investigation, A.I.; resources, A.A.; data curation, A.N.A.-T.; writing—original draft preparation, A.A. and A.A.-A.A.-r.; writing—review and editing, A.N.A.-T. and A.I.; visualization, M.M.; supervision, A.A.; project administration, A.A.-A.A.-r.; funding acquisition, A.A. All authors have read and agreed to the published version of the manuscript.

**Funding:** This research received no external funding.

**Institutional Review Board Statement:** Not applicable.

**Informed Consent Statement:** Not applicable.

**Data Availability Statement:** Not applicable.

**Conflicts of Interest:** The authors declare no conflict of interest.

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
