# Peer review of "Sustaining Organizational Outcomes in Manufacturing Firms: The Role of HRM and Occupational Health and Safety"

_sustainability, doi:10.3390/su16031035_

Round 1

Reviewer 1 Report

I extend my sincere thanks to the researchers for giving me this wonderful opportunity to read this article. This scientific paper has amazing results, but it needs a few modifications before it is approved for publication.

Abstract:

1-    It must not include an introduction.

2-    The key words They must be arranged either alphabetically or according to their order in the study model.

3-    The circumstances of collecting data from the sample were not explained. It is better to show this clearly, and this is more important than providing an introduction and losing the number of words allowed, for example, the size of the population, how to determine the sample size, or the target geographical area.

4-    The researchers did not show the Practical Implications or originality of the study’s results in the introduction or the limitations. Please work to shorten them in the abstract.

5-    The contribution is also not completely clear. There are many studies that have dealt with the subject of human resources management and occupational health and safety, and you mentioned that in more than one place in the study.

Introduction part:

1-    Line 95. you indicated that there are some studies that have addressed the issue of the relationship between human resources management and occupational health and safety, but the gap research between the current study and previous studies has not been shown.

2-    In the introduction section, I would like to see the author highlight the research gap and research problem, which is one of the important parts to justify the research significance.

3-    Using abbreviations for the first time must be accompanied by a definition, and there is no objection to using the abbreviation alone later.

 In the hypotheses part:

1-    In the first hypothesis, the study population was not specified, but in the rest of the hypothesis, the study population was specifically specified (industrial companies).

2-    Also, in the third hypothesis, you used abbreviations, while in the fourth hypothesis, you used words and abbreviations. I hope to work on unifying the hypotheses in the same way, unless the difference is intentional, and I certainly do not think so.

Sample section

1-    You used a very old reference Krejcie and Morgan 1970, Are there no better modern statistical references than this one?  very old reference and the scientific sources are full.

2-    It is not specified how the sample size is calculated

3-    you did not provide any proof that this type of sample selection method is appropriate and valid for industrial companies.

4-    Here also, the circumstances of selecting the sample were not shown or explained. 156 questionnaires were distributed... Are all of them suitable for statistical analysis??

In the results section:

-        The experience factor was divided into two categories: less than ten years and more than ten years. If one of the respondents had 10 years of experience, which of the categories would he choose?

In conclusion section:

-        please comment If there are any objective errors, or if the conclusions are not supported, you should detail your concerns.

References section:

Very good, but there are some incomplete references, such as the number of pages or volume.

Author Response

We deeply appreciate the time and effort invested by you in evaluating our article. Your insights and feedback are invaluable to the quality and rigor of our research.

Abstract:

Point 1.  It must not include an introduction.

Response to point 1: Thank you for this comment, the introduction was deleted, and only the first two lines were kept to show research gaps.

Point 2. The keywords must be arranged either alphabetically or according to their order in the study model.

Response to point 2: Thank you so much for this comment, the authors rearranged the keywords according to their order in the title.

Point 3. The circumstances of collecting data from the sample were not explained. It is better to show this clearly, and this is more important than providing an introduction and losing the number of words allowed, for example, the size of the population, how to determine the sample size, or the target geographical area.

Response to point 3:  This is a very valuable comment, the authors add methods for data collection (a cross-sectional survey, using a convenience sampling technique, and a web-based form). Also, specify the target company and geographical area.

Point 4. The researchers did not show the Practical Implications or originality of the study’s results in the introduction or the limitations. Please work to shorten them in the abstract.

Response to point 4:  The authors revised the abstract and provided a novelty contribution of the study to the literature by uncovering the intricate interplay of HRM, OHS, and SOO.

Point 5. The contribution is also not completely clear. There are many studies that have dealt with the subject of human resources management and occupational health and safety, and you mentioned that in more than one place in the study.

Response to point 4:  previous studies that have dealt with the subject of human resources management and occupational health and safety, nevertheless didn’t uncover their impact on sustainable outcomes and also didn’t examine the mediating role of OHS between HRM and sustainable outcomes. Therefore, this study contributes to the literature by attempting to fill these research gaps.

Introduction part:

Point 1:   Line 95. you indicated that there are some studies that have addressed the issue of the relationship between human resources management and occupational health and safety, but the gap research between the current study and previous studies has not been shown.

Response to point 1:  The Authors mentioned that previous studies investigated the relationship between HRM and OHS, while the effect of OHS on sustainable outcomes was rarely examined. Furthermore, the mediating role of OHS between HRM and sustainable outcomes has not yet been explored. In addition, the gaps in the region such as the Middle East, and industry also mentioned in lines 97 to 105.  

Point 2: In the introduction section, I would like to see the author highlight the research gap and research problem, which is one of the important parts to justify the research significance.

Response to point 2: In the introduction section, The Authors highlighted the research gap and research problem, and highlighted the mediating role of OHS between HRM and sustainable outcomes has not yet been explored. In addition, the gaps in the region such as the Middle East.

Point 3: Using abbreviations for the first time must be accompanied by a definition, and there is no objection to using the abbreviation alone later.

Response to point 2: Thank you for this comment.

In the hypotheses part:

Point 1: In the first hypothesis, the study population was not specified, but in the rest of the hypothesis, the study population was specifically specified (industrial companies).

Response to point 1: The authors add the study population to the first hypothesis, (industrial companies).

Point 2: Also, in the third hypothesis, you used abbreviations, while in the fourth hypothesis, you used words and abbreviations. I hope to work on unifying the hypotheses in the same way, unless the difference is intentional, and I certainly do not think so.

Response to point 2: The Authors unify the hypotheses (from 1 to 4) by using the same way (words and abbreviations)

Sample section

Point 1: You used a very old reference Krejcie and Morgan 1970, Are there no better modern statistical references than this one?  very old reference and the scientific sources are full.

Response to point 1: the author used the reference Krejcie and Morgan 1970, Even though it is old, nevertheless is still used in modern studies. However, besides that, we used a modern program to calculate the sample size.

Point 2: It is not specified how the sample size is calculated.

Response to point 2:  sample size was calculated by the table of sample size by Krejcie and Morgan 1970, in addition, a modern program was used and attached screenshot of the result. 

Point 3: you did not provide any proof that this type of sample selection method is appropriate and valid for industrial companies.

Response to point 3:  this type of sample selection method is appropriate and valid for any organization in any sector.

Point 4: Here also, the circumstances of selecting the sample were not shown or explained. 156 questionnaires were distributed... Are all of them suitable for statistical analysis??

Response to point 4:  The authors used Google Forms, which has a mandatory answer feature for all questions, and thus, there is no missing data or incomplete survey.

In the results section:

Point 1: The experience factor was divided into two categories: less than ten years and more than ten years. If one of the respondents had 10 years of experience, which of the categories would he choose?

Response to point 1: The experience factor was divided into two categories: less than ten years and more than ten years. There is no choice for 10 years, just less or more than 10 years. If the respondents have experienced less than 10 years by one day, so he/she in the category of less than 10 years. and if He/she has experienced more than 10 years by one day, so he/she is in the category of more than 10 years.

The Authors take this operationality to express their appreciation for your time and valuable advice.

Reviewer 2 Report

ID Sustainability-2638256. Thanks for giving me the chance to review a manuscript about the Sustaining Organizational Outcomes in the manufacturing firms: The role of HRM and Occupational Health and Safety. This research is a hot topic globally and is well written. The discussion and analysis are also very scholarly, highlighting Structural Equation Modelling and used Smart-PLS. But there is some correction, please consider my comments and suggestions.

I advise the author to make minor changes in order to eliminate grammar roughness and improve the body text's imperfection.

What is the theoretical contribution of this research? Can you compare your findings with the previous studies?

The relevance of the model should be analyzed. I'm particularly in doubt regarding the Assessment of Measurement Model, which apparently does not provide any value or added information for the article. besides being from an external source.

The conclusion is particularly poor. My recommendation is to create at least two subsections, one for the theoretical, managerial, or other types of implications. And one last subsection for limitations and future research directions.

 Good Luck!

minor changes in order to eliminate grammar roughness

Author Response

Response to Reviewer 2 Comments

We deeply appreciate the time and effort invested by you in evaluating our article. Your insights and feedback are invaluable to the quality and rigor of our research.

Point 1: This research is a hot topic globally and is well-written.

Response to point 1:  thank you so much for this comment, we really appreciate your valuable time.

Point 2: The discussion and analysis are also very scholarly.

Response to point 2:  thank you so much for this comment, I do appreciate your valuable time.

Point 3. I advise the author to make minor changes in order to eliminate grammar roughness and improve the body text's imperfection.

Response to point 2:  The researchers made a minor correction in the body of the text, and checked for grammar mistakes.

Point 3: What is the theoretical contribution of this research? Can you compare your findings with the previous studies?

Response to point 3:  Thank you for this comment, the researchers added one section about the theoretical contribution of this research (lines 585 to 594).

Point 4: The relevance of the model should be analyzed. I'm particularly in doubt regarding the Assessment of Measurement Model, which apparently does not provide any value or added information for the article. besides being from an external source.

Response to point 4:  In the context of our study, the analysis of the measurement model primarily serves to demonstrate the reliability, divergent, and discriminant validity of the model. This ensures that the constructs used are consistent, and distinct, and accurately measure the intended variables. It's a critical step to validate the robustness of our findings before delving deeper into the structural relationships.

Point 5: The conclusion is particularly poor. My recommendation is to create at least two subsections, one for the theoretical, managerial, or other types of implications. And one last subsection for limitations and future research directions.

Response to point 4:  Thank you for this comment, the Authors added two subsections, one for the theoretical implication and one for the practical implication, in addition, the Authors added one more subsection for limitations and future research directions.

Point 5: minor changes in order to eliminate grammar roughness.

Response to point 5:  The Authors made some minor corrections for some grammar mistakes.

The Authors take this operationality to express their appreciation for your time and valuable advice.

Reviewer 3 Report

The manuscript is well-structured, presenting the subject matter concisely with clear contributions to the field. While the authors cited relevant literature that supports the empirical data, specific citations are needed for the assertion on lines 95 to 97. Since this statement emphasizes the importance of this study, it should be built upon solid literature. Additional references for mental health and well-being are also required, as these constructs are complicated. 

The theoretical background is original, with a commendable effort to merge previous research with current perspectives on the topic. Overall, the authors presented the hypotheses in a logical order with supporting literature. However, the research question must be clearly articulated to set the stage for such hypotheses.

The results are clear and enriched with relevant graphs and tables, facilitating a better understanding of the study's empirical findings. However, attention to the quality of figure presentations, specifically improvements in Figures 5 and 6, and correction of any skewed representations, such as in Figure 1, are recommended. While these enhancements may not directly impact the scholarly merit of the work, they are crucial for the audience's comprehension of the manuscript.

I enjoyed reading the manuscript.

The quality of written English is commendable, clear and concise. Minor grammatical inconsistencies could be refined to enhance the manuscript's readability further.

Author Response

We deeply appreciate the time and effort invested by you in evaluating our article. Your insights and feedback are invaluable to the quality and rigor of our research.

Point 1. The manuscript is well-structured, presenting the subject matter concisely with clear contributions to the field.

Response to point 1:  thank you so much for this comment, we really appreciate your valuable time.

Point 2: While the authors cited relevant literature that supports the empirical data, specific citations are needed for the assertion on lines 95 to 97. Since this statement emphasizes the importance of this study.

Response to point 2:  The Author added citations in lines 95 to 100 to emphasize the importance of this study.

Point 3: Additional references for mental health and well-being are also required, as these constructs are complicated.

Response to point 3: The researcher provides related references to the importance of employees’ mental health and well-being (please see references 15, 16, and 17). Lines 81 to 85.

Point 4: The theoretical background is original, with a commendable effort to merge previous research with current perspectives on the topic.

Response to point 4: thank you so much for this comment, we really appreciate your valuable time.

Point 5: Overall, the authors presented the hypotheses in a logical order with supporting literature.

Response to point 4: thank you so much for this comment, we do appreciate your valuable time.

Point 6: The results are clear and enriched with relevant graphs and tables, facilitating a better understanding of the study's empirical findings.

Response to point 6: thank you so much for this comment, we really appreciate your valuable time.

Point 7: attention to the quality of figure presentations, specifically improvements in Figures 5 and 6, and correction of any skewed representations, such as in Figure 1, are recommended. While these enhancements may not directly impact the scholarly merit of the work, they are crucial for the audience's comprehension of the manuscript.

Response to point 6: due to the figures taken by a screenshot from the outcomes of the program, the Authors used IA and improved the quality of the figures.

The Authors take this operationality to express their appreciation for your time and valuable advice.